# Transmembrane Chloride Intracellular Channel 1 (tmCLIC1) as a Potential Biomarker for Personalized Medicine

**DOI:** 10.3390/jpm11070635

**Published:** 2021-07-05

**Authors:** Francesca Cianci, Ivan Verduci

**Affiliations:** Department of Biosciences, University of Milan, 20133 Milan, Italy; ivan.verduci@unimi.it

**Keywords:** CLIC1, tmCLIC1, tumor, neurodegeneration, biomarker, personalized medicine

## Abstract

Identification of potential pathological biomarkers has proved to be essential for understanding complex and fatal diseases, such as cancer and neurodegenerative diseases. Ion channels are involved in the maintenance of cellular homeostasis. Moreover, loss of function and aberrant expression of ion channels and transporters have been linked to various cancers, and to neurodegeneration. The Chloride Intracellular Channel 1 (CLIC1), CLIC1 is a metamorphic protein belonging to a partially unexplored protein superfamily, the CLICs. In homeostatic conditions, CLIC1 protein is expressed in cells as a cytosolic monomer. In pathological states, CLIC1 is specifically expressed as transmembrane chloride channel. In the following review, we trace the involvement of CLIC1 protein functions in physiological and in pathological conditions and assess its functionally active isoform as a potential target for future therapeutic strategies.

## 1. CLIC Proteins, a Focus on CLIC1

Chloride channels exert different functions in every stage. They are involved in ion homeostasis, fluid transportation, regulation of cell volume, cytoskeletal rearrangement, and cellular motility. An example is the interplay between swelling-activate K+ and Cl- channels during volume regulation due to cellular hypertonicity. In this case, both channels cooperate allowing a net efflux of ions. Chloride Intracellular Channel proteins (CLIC) are a class of proteins both soluble and integral membrane forms. Littler et al. showed CLIC proteins to be highly conserved in chordates with the presence of six vertebrate paralogues [1]. CLIC proteins diverge from the canonical ion channel structure. Using phylogen and structural techniques they found CLIC proteins exert enzymatic functions. CLIC proteins show 240 aminoacidic sequence belonging to the glutathione S-transferase (GST) fold superfamily. This suggests a well-conserved intracellular activity playing a role in redox balance. In particular, the enzymatic active site was shown to be the Cys24, distinct from canonical GSTs which has been shown to exhibit a thiol group [2]. As a variety of metamorphic proteins, CLIC proteins undergo important changes in structure switching from the globular hydrophilic form to the transmembrane hydrophobic structure. It was found that the membrane insertion is elicited by oxidizing conditions and pH changes [2,3,4,5]. Oxidation-dependent insertion is likely to be mediated by the active residue Cys24 as the C24A mutation alters the redox sensitivity of the channel and—for CLIC1—its electrophysiological characteristics. CLICs channel activity at physiological pH has been shown to be minimal while the presence of membrane-expressed rises with a change in pH [4]. To date, little is known about the function of CLIC proteins in native tissues. An example is the enrichment of CLIC4 and CLIC5 in the human placenta, in particular the apical section of microvilli in trophoblasts. Jentsch et al. propose this phenomenon to be secretion dependent [6]. Given that secretory vesicles are known to establish an acidic pH for the correct assembly, this may need a chloride conductance to balance the proton transport coupled to vesicle acidification. At the same time, cytoplasmic acidification may be a switch for their membrane insertion. Significant expression of mRNAs encoding CLIC1, 2, 4, and 5 were found to be expressed in human hepatocellular carcinoma and metastatic colorectal cancer in the liver. CLIC2, in particular, was predominantly expressed in non-cancer tissues surrounding cancer masses where may be involved in the formation or maintenance of tight junctions which allows the intravasation of cancer cells to form metastasis [7]. Moreover, the protein was found to be a promising biomarker for efficacy of treatment of advantage-stage breast cancer. It was found to be co-expressed with PD-1 and PD-L1 and its increased expression was associated with a favorable prognosis and enrichment of multiple tumor-infiltrating lymphocyte types, particularly CD8+ T cells [8]. CLIC2 mutation were reported also to be associated with intellectual disability. Large-scale next generation resequencing of X chromosome genes identified a missense mutation in the CLIC2 gene on Xq28 in a male with X-linked intellectual disability and not found in healthy individuals. In particular, point mutation p.H101Q seems to be fundamental for membrane insertion of CLIC2 protein, suggesting that p.H101Q may be a disease-causing mutation, the first correlated with CLIC superfamily [9]. As for CLIC1 and CLIC2 proteins, also CLIC3 was found to be overexpressed in bladder cancer [10], pancreatic cancer [11], and metastatic breast cancer [12]. Its role in carcinogenesis was demonstrated to be correlated with the activity of CLIC1 as glutathione-dependent oxidoreductase activity that drive angiogenesis and increases invasiveness of cancer cells with transglutaminase-2 in ovarian cancer [13]. CLIC4 protein were associated to several normal cellular functions as the recruitment of NLRP3 complex during the formation of inflammasome [14], and—together with CLIC1 function and ezrin—they bridge plasma membrane and actin cytoskeleton at the polar cortex and cleavage furrow to promote cortical stability and successful completion of cytokinesis in mammalian cells [15]. CLIC proteins were found to also be involved in endothelium architecture. Tavasoli and colleagues have demonstrated that both CLIC1 and CLIC5A activate ezrin, radixin, and moesin proteins for the formation of the lumen of vessels. As a result, the dual CLIC4/CLIC5-deficient mice developed spontaneous proteinuria, glomerular cell proliferation, and matrix deposition [16]. Among CLIC proteins, despite CLIC1 is considered a chloride channel, its RNA transcripts lack the specific sequences for membrane insertion. Thus, it does not follow secretary pathways typical of canonical ion channels through endoplasmic reticulum and Golgi apparatus [17]. For this reason, to demonstrate that CLIC1 protein is a channel, rather than a modulatory element, recombinant protein inserted in artificial lipid bilayers has been used. These experiments evidenced a conductance in internal–external 140 mM KCl of 30 pS with a variety of substates suggesting that the integral membrane form of CLIC1 would comprise four single monomers [2,3,5,17,18]. Interestingly, each monomer is able to allow an anionic flux. This feature is supported by experiments on lipid bilayers performed by Tulk et al., showing that at low protein concentration there is a linear relationship between protein concentration and channel activity while reaching a saturation at higher concentration [4]. These experiments show that there is an assembly of single CLIC1 monomers at the membrane level and not prior the insertion. A similar behavior was observed by Warton et al. using the Tip-Dip technique. In particular, the authors show that an initial small conductance with slow kinetics is replaced by a high conductance fast kinetics determining four times the initial conductance recorded, equating WT CLIC1 channel [5]. The authors concluded that the high conductance fast kinetics represent the final step of single monomers membrane insertion and subsequent aggregation and cooperation. Using patch clamp technique on CHO-K1 cells overexpressing CLIC1, Tonini et al. were able to measure CLIC1 conductance both in cellular and nuclear membrane. They showed an evident sensitivity and selectivity to chloride concentrations. Furthermore, taking advantage of inside-out and outside-out patch clamp configurations, they concluded that the N-terminus domain projects outside of the cell, while the C-terminus is directed inwardly [19]. More recently, taking advantage of tethered bilayer lipid membranes combined with impedance spectroscopy, it has been shown that redox environment regulates CLIC1 membrane insertion but also that cholesterol represent an initial membrane binding site which would favor subsequent transmembrane protein rearrangements or oligomerizations [20].

## 2. CLIC1 in Non-Pathological States

In 2009, Qiu et al. published that, although the deletion of *Clic1* gene in mouse did not cause embryonic lethality, mice developed a mild bleeding disorder with a higher platelet number and longer bleeding times compared to control group [21]. The authors propose this mechanism to be related to platelet P2Y12 receptor. Despite the mechanism of action relating CLIC1 to P2Y12R is still unknown, the authors suggest that the redox balance could be involved in the pathway. This conclusion is supported by the observation that P2Y12R requires free thiol groups (Cys17 and Cys270) to elicit its function. In this picture, CLIC1 would participate in the control of redox environments which, in turn, may induce platelet activation [21]. More recently, relatively abundant levels of CLIC proteins have been found in human bronchial epithelial cells primary cultures. In particular, it has been shown CLIC1 to be able to modulate cAMP-induced chloride currents [22]. The authors show that one of the well-known CFTR antagonists on the market, PPQ-102 abolished the currents elicited by isoproterenol or forskolin. Despite the fact that this effect is attributed, usually, to the activation of CFTR, RNA Seq datasets show the absence of this transcript in the analyzed cells. As a matter of fact, knockdown of CLIC1 inhibits significantly cAMP-induced chloride currents, suggesting a direct contribution of CLIC1 to this chloride conductance. The authors concluded that manipulating CLIC1 properties may enhance Cl- conductance, thus proposing the possibility to potentiate CLIC1 functional activity as a possible therapeutic strategy in conditions of impaired chloride homeostasis as cystic fibrosis [22].

## 3. Activation of Transitory Allostasis through CLIC1 Function

In tissues, CLIC1 is found mainly as a cytoplasmic protein able to shuttle to plasma membrane following a remodeling elicited by perturbation of cellular homeostasis. The two major players are oxidative stress and pH alterations. Overproduction of reactive oxygen species (ROS), modifies the structure of the protein through the formation of a disulphide bridge between Cys24 and Cys59, considered the essential oxidoreductase residues, which promotes membrane docking [3,20,23]. Transient oxidative stress is proper of physiological functions as cell cycle progression [24] and innate immune responses [25]. As a matter of fact, CLIC1 functional activity was demonstrated to be increased in active dividing cells, or cells that have completed mitotic processes [5,26]. Recently, it has been demonstrated a correlation between CLIC1 activity and Peroxiredosin 6 (Prdx6), a protein that influences redox environment. Downregulation of Prdx6 causes an alteration of oxidation of CLIC1 GST domain, causing cell swelling and cell cycle arrest [27]. Redox modifications are also involved in immune functions. Valenzuela and colleagues cloned CLIC1 for the first time to investigate its role in activated macrophages, where it was found to be overexpressed compared to resting immune cells [17]. In resting macrophages CLIC1 is cytoplasmic, while after stimulation with pro-inflammatory factors, CLIC1 migrates to the plasma membrane of cells and colocalizes with the NADPH oxidase complex subunit Rac2 [28]. The transient translocation of CLIC1 in plasma membrane of macrophages promotes phagosome acidification from 4.4 to 3.4. *Clic1^−/−^* macrophages display impaired phagosome proteolysis and altered ability to kill microorganisms and to expose antigens to T cells [28]. Moreover, it has been demonstrated that tmCLIC1 activity contributes to the formation of inflammasome modulating NLRP3 complex [14,25], a multiproteic complex crucial for the immune system, which regulates the activation of Caspase 1 and release of cytokines as Interleukine 6 (IL-6) and Interleukine 8 (IL-8), promoting pyroptosis (inflammatory programmed cell death). In addition, Domingo-Fernandez and colleagues have found CLIC1-mediated current essential in enzymatic cascade for activation of NLRP3 inflammasome. NLRP3 agonist (NEK7) causes a potassium efflux that provokes mitochondrial damage and release of ROS with a consequent activation of Caspase 1 and release of IL-6 and IL-8 [14,25]. Recent studies have revealed a role of CLIC1 protein in angiogenesis supporting ROS production for proliferation and migration of endothelial cells [29,30,31]. Tung and Kitajewski demonstrate that CLIC1 plays a role in endothelial cell growth, sprouting, branching, and migration regulating Integrins subunits β1 and α3 expression [32]. Moreover, CLIC1 contributes to endothelial damage responses cascade translocating in plasma membrane of human umbilical vein endothelial cells (HUVEC) where it enhances the expression of tumor necrosis factor α (TNF α), IL-1ß, intracellular adhesion molecule 1 (ICAM1), and vascular cell adhesion protein 1 (VCAM1) [33]. The mechanism of action by which CLIC1 could support the activation of transitory allostatic processes was investigated by Milton and colleagues that postulated a feed forward mechanism involving CLIC1 protein functional activity. NADPH oxidation causes an extrusion of electrons throughout plasma membrane and the resulting depolarization downregulates the bioenergy of enzymatic activity. CLIC1 conductance during ROS overproduction supports the setting of the resting membrane potential, allowing electrogenic activity of enzymes and ensuring further ROS production. CLIC1 translocation to plasma membrane could be considered a compensative mechanism to support cellular physiological functions in the presence of oxidative stress [34]. Considering these evidence, it has been postulated that CLIC1 acts as a second messenger which can translocate transiently, as in macrophages activation and cell cycle phases, or chronically to the plasma membrane of hyperactivated systems [26]. Chronic oxidative stress and pH alkalization are hallmark for different pathologies. Therefore, chronic expression of CLIC1 in plasma membrane could be considered as a promising target to identify pathological conditions.

## 4. CLIC1 during Chronic Allostasis

Literature reports CLIC1 to be linked to cancer development and neurodegenerative processes. Both share a critical modification of the overall oxidative state towards an unbalance of reactive oxygen species, despite they are associated to opposite outcomes.

### 4.1. CLIC1 in Solid Tumors

CLIC1 protein was found to be overexpressed in several solid tumors. According to “The Human Protein Atlas” RNA database, gliomas, colorectal cancer, lung, ovarian, pancreatic, prostate, breast, and melanoma cancers have shown higher levels of CLIC1 RNA. Although the role of CLIC1 protein in tumorigenesis is still unclear, in recent years, different studies have elucidated the possible involvement of CLIC1 in tumor formation and progression. Lu and colleagues have postulated that CLIC1 acts as an oncogene in pancreatic cancer. Here, patients with CLIC1-positive tumors have demonstrated worse overall survival compared to those with CLIC1-negative tumors [35]. CLIC1 expression is correlated to a poor prognosis not only in pancreatic cancer, but also in tumors as lung cancer [36], ovarian cancer, where CLIC1 upregulation was correlated to chemotherapy resistance [37], gallbladder, and gastric cancers [38], where it was found to promote cells proliferation via MAPK/AKT regulation [39] and facilitates the formation of tumor-associated fibroblasts [40]. In addition, CLIC1 protein upregulation correlates with the level of aggressiveness and metastatic potential of colorectal cancer cells [41], where it was demonstrated to regulate cell volume and ROS level. One of the aspects of cancer is its heterogeneity, it is composed of differentiated cells, representing the major component of the tumor mass, and cancer stem cells (CSCs) [42]. Despite cancer stem cells constitute a small percentage of tumor cells (0.05–1%), they are the prime sources of tumor recurrence and metastasis as they confer resistance to chemo and radiotherapies [43]. Cancer stem cells are able to take advantage of the aberrant redox system. In particular, it was demonstrated that oxidative stress and gene-environment interactions support the development of a huge variety of solid tumors, as glioblastoma, breast, prostate, pancreatic, and colon cancer [44]. CLIC1 expression in plasma membrane is strictly dependent on redox homeostasis and pH levels supporting tumorigenesis and development of cancer [26]. Therefore, considering the role of oxidative stress in CSCs, a hypothesis could be that tmCLIC1 protein has an important role in CSCs physiology. In particular, CLIC1 functional activity was assessed in glioblastoma stem cells (GSC). Electrophysiological experiments have revealed a significant increase in CLIC1-mediated current in cells positive to stem/progenitor cell markers (Sox2, Nestin), demonstrating that tmCLIC1 is chronically expressed in GSCs compartment compared to differentiated ones [45]. In addition, Setti and colleagues have shown that tmCLIC1 functional activity has a pivotal role in glioblastoma stem cells (GSC), supporting self-renewal. Moreover, the tumorigenic capability of GSCs, in which CLIC1 is inhibited or silenced, was significantly reduced [42,46]. tmCLIC1 protein was demonstrated to have a pivotal role in cell cycle progression and cellular proliferation promoting G1/S cell cycle transition in GSCs. Peretti and colleagues proposed tmCLIC1 to contribute to a feed forward mechanism together with NHE1 proton pump and NADPH oxidase, promoting ROS overproduction [26]. According to these evidences, tmCLIC1 could represent an important target to sensitize CSCs toward chemo and radiotherapies, increasing tumor response to anticancer treatments.

### 4.2. CLIC1 in Neurodegenerative Processes

The first evidence about tmCLIC1 involvement in neurodegenerative processes dates to 2004 [47]. In the paper, the authors show an increase in protein expression, and enhanced functional activity in response to Aβ peptide incubation in microglial cells. In addition, the Aβ-induced release of Nitric Oxide (NO), as well as TNF-α is prevented by impairing tmCLIC1 function by its pharmacological inhibition and by RNA interference, supporting an active role of the protein during the neurodegenerative processes. A possible mechanism by which tmCLIC1 would support the neurodegenerative process was suggested in a following work [34]. In particular, its ion channel activity would sustain the Aβ-induced ROS production by microglial cells, acting as a charge compensator, which balances the electrogenic activity of the NADPH Oxidase (NOX2). The correlation between tmCLIC1 functional activity and neurodegeneration was further confirmed by the observation of high protein levels in section from 3xTg-AD mouse brain, characterized by a progressive deposition of Aβ during their life. The authors concluded that tmCLIC1 inhibition could represent a promising anti-inflammatory target for Alzheimer’s Disease therapy. This was further strengthened by the work of Paradisi et al. [48] where the authors have shown that the inhibition of tmCLIC1 activity by pharmacological blockers and the suppression by RNA interference does not alter the activation of resident microglia. In particular, they show that affecting tmCLIC1 ion channel produces a reduction in some toxic aspects of activated microglia (i.e., NO production) while leaving unaltered its phagocytic ability. Moreover, in a following work it was shown that by co-culturing neurons and microglia in a two-chamber transwell, IAA94 treatment was able to prevent the neurotoxicity induced by the treatment with Aβ1-42 peptide [49]. In addition, they show that tmCLIC1 ion channel blocker IAA94 failed to protect cortical neurons when directly exposed to Aβ1-42 suggesting the phenomenon to be dependent on the lack of CLIC1 expression by these cells, or lack of a pro-oxidant environment to provoke CLIC1 membrane insertion. Since soluble and fibrillar Aβ are both able to induce ROS production in microglia it is reasonable to think that this is the cellular subtype where tmCLIC1 inserts forming a functional channel. In a recent study it was shown that peripheral blood mononuclear cells (PBMC) and, in particular, monocytes, isolated from Alzheimer’s Disease patients show an overexpression of CLIC1 mRNA that is accompanied by a significative increase in transmembrane protein [50]. The authors concluded that the study could pave the way for future developing strategies aimed at discriminating between healthy and patients with ongoing neurodegenerative processes. A similar result was achieved by Miller et al. [51] by means of blood microarray data and machine learning approach to predict the cognitive status using three groups: normal cognitive, mild cognitive impairment, and probable Alzheimer’s Disease. Blood RNA levels showed that CLIC1 was the only significant probe to change among groups although, to date, it cannot be sufficient to predict neurodegenerative progression throughout Alzheimer’s Disease.

## 5. Conclusions

CLIC1 protein has been described in several papers associated to different pathological states, from neurodegenerative processes to solid tumors. However, CLIC1 is better described as a determinant of adaptation in different biological contexts, rather than being able to induce disease development. CLIC1 is one of the several proteins that react to stress conditions promoting cell survival. CLIC1 protein is conserved from yeast to humans and expressed in the cytoplasm of every cell types. A peculiar feature of CLIC1 is to be present on the cell membrane only in hyperactivated systems. This is particularly important for cells targeting in diagnosis of a pathological state or as a possible pharmacological target in therapeutic protocols. In all the living organisms, cells usually maintain a dynamic state of equilibrium defined as homeostatic state. In case of profound and endless changes in the surrounding environment, there are two main possible cell reactions. Cells involved in defense mechanisms, as cells of the immune system, become persistently activated. On the contrary, most somatic cells are not able to cope with the new conditions and are negatively selected. However, a small percentage of somatic cells are able to activate stable allostatic mechanisms and survive. It is important to underline that allostatic cells are not similar to the original ones, but they form a new population with completely different physiological characteristics. The majority of these new cell populations represent an adaptation (Figure 1).

From a physiological point of view, tumorigenesis can be defined as an adaptation. Uncontrolled proliferation can be seen as an extreme survival mechanism in response to a chronic stress state generated by a hostile environment. In this scenario, among other proteins, CLIC1 is instrumental for this transformation. In particular, as a metamorphic protein, its translocation to the plasma membrane could be a first line response to diverse stimuli. Transmembrane insertion and the consequent increased ionic permeability can represent a fast reaction to counteract cytoplasmic unbalance conditions. As mentioned before, cytoplasmic CLIC1 protein binds GSH. In adverse conditions, the rearrangement of the protein structure releases GSH, contributing to buffer cytoplasmic oxidation. At the same time, CLIC1 association with an ionic channel guarantees the possibility to disperse any excess of charges that could accumulate during a persistent condition of oxidative stress. In the central nervous system, the presence of high concentration of β-amyloid generates a state of oxidative stress that triggers microglia activation with the intent of phagocyte oligomers and fibrils. The release of a huge amount of amyloid causes microglia hyperactivation that becomes harmful for neurons and glial cells, generating the process of neurodegeneration. In solid tumors, the process may be different. Chronic conditions of oxidative stress due to environmental causes like pollution, smoke, wrong or excessive alimentation, or hormonal dysregulation could induce cell “adaptation” giving rise to a neoplastic process. Both these conditions show a direct correlation between the presence of membrane CLIC1 and the state of microglia activation or tumors aggressiveness. The initial mechanism of neurodegeneration and tumorigenesis in sporadic cases is still obscure. However, CLIC1 protein functional expression as a membrane protein could be symptomatic of hyperactivated states as that of neurodegeneration or the neoplastic transformation. In this perspective, tmCLIC1 could represent a promising element as a diagnostic tool, as well as a therapeutic target.

## Figures and Tables

**Figure 1 jpm-11-00635-f001:**
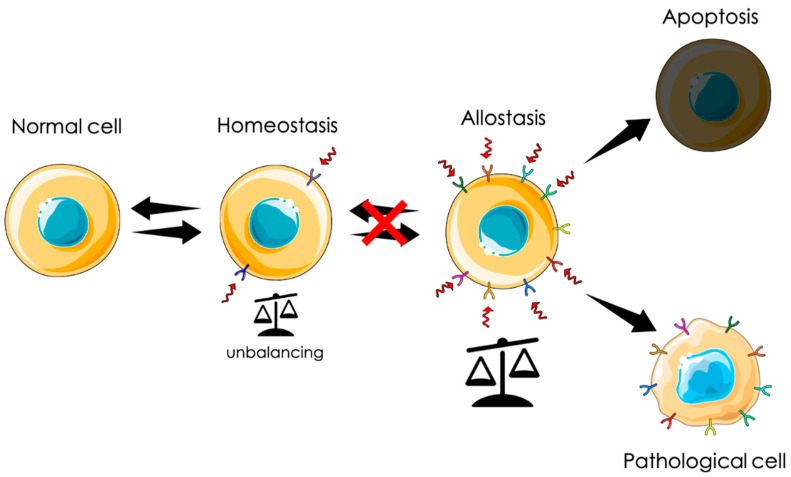
Graphic representation of the dual behavior of cells under chronic stress. All cells constantly rely in a dynamic state of equilibrium defined as homeostasis, in which cells respond to stress stimuli with a transient activation of allostatic mechanisms. When the stimuli became persistent, the hyperactivation is irreversible. In a major part of the cases, this is not compatible with cell life, leading to cell death. In rare cases, the hyperactivation has, as a consequence, the establishment of a new steady state far different from the initial cell.

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
