# Peer review of "Transmembrane Chloride Intracellular Channel 1 (tmCLIC1) as a Potential Biomarker for Personalized Medicine"

_jpm, 2021, doi:10.3390/jpm11070635_

Round 1
Reviewer 1 Report
This paper reviews the role of chloride channels in tumor formation and Alzheimers disease. The figures accompanying the manuscript are not informative and should be deleted. The review needs substantial editing and rewritting as it tends to ramble with a lot of unnecessary words. I suggest the authors begin by reviewing the structure and known physiology of these chloride channels before launching into more speculative discussion of pathophysiological roles. For example, I suggest a more compact, but informative discussion of astroglial ion transport mechanisms in relation to beta amyloid metabolism. Overall, there are some interesting points here but it is a difficult paper to read and points are not clear
Author Response
This paper reviews the role of chloride channels in tumor formation and Alzheimers disease. The figures accompanying the manuscript are not informative and should be deleted.
Dear Colleague,
Thanks for your critic and well-explained comments. Now, we removed 2 figures. The only left is the one concerning our hypothesis about CLIC1 activation in chronic allostasis. We find that it is useful for better understanding our assumptions.
The review needs substantial editing and rewritting as it tends to ramble with a lot of unnecessary words.
We think that this comment is referred most to the paragraph named as “Activation of transitory allostasis through CLIC1 function”. We tried to rephrase and remodulate it. Although this can be considered complex, we believe it as a crucial passage to understand the peculiar feature of CLIC1 protein.
I suggest the authors begin by reviewing the structure and known physiology of these chloride channels before launching into more speculative discussion of pathophysiological roles.
We tried to review the structure of CLIC1 channel in the first paragraph despite little is known about its structure as a membrane player due to its peculiar features and its sensitivity to cytoplasmic perturbation. This makes difficult its isolation in the transmembrane form.
For example, I suggest a more compact, but informative discussion of astroglial ion transport mechanisms in relation to beta amyloid metabolism.
We discussed tmCLIC1 implication in neurodegeneration and beta amyloid-induced microglia activation, including ion transportation and its mechanism of action as a putative charge compensator.
Overall, there are some interesting points here but it is a difficult paper to read and points are not clear.
Reviewer 2 Report
Dear Autors,
I have read your review well. I think it is well-organized and explained in an easy-to-understand manner even for those who are not familiar with this field.
* Major
The CLIC1 is that many researchers have fully explained the role of CLIC1 in cancer experiments and are using it in many fields. 1, 2, 3. Also, it has been highlighted in recent review journals. 4 It's hard to find the merits of this review.
However, I think it is an advantage to focus on CLIC1 and deal with it in more depth.
* Minor
It would be good if the functions and roles of CLIC2~6 were schematically added and added simply.
- Ref
- Anticancer Research December 2020, 40 (12) 6879-6884; DOI: https://doi.org/10.21873/anticanres.14710
- SciEntific Reports | (2018) 8:14725 | DOI:10.1038/s41598-018-32885-23.
- Onco Targets Ther. 2018; 11: 8073–8081. Published online 2018 Nov 12. doi: 10.2147/OTT.S181936
- Front. Physiol., 14 February 2020 | https://doi.org/10.3389/fphys.2020.00096
Author Response
Dear Autors,
I have read your review well. I think it is well-organized and explained in an easy-to-understand manner even for those who are not familiar with this field.
* Major
The CLIC1 is that many researchers have fully explained the role of CLIC1 in cancer experiments and are using it in many fields. 1, 2, 3. Also, it has been highlighted in recent review journals. 4 It's hard to find the merits of this review.
However, I think it is an advantage to focus on CLIC1 and deal with it in more depth.
* Minor
It would be good if the functions and roles of CLIC2~6 were schematically added and added simply.
Dear Colleague,
Thanks for your comments. Although we have mentioned to CLIC1-6 proteins in the first paragraph of this review, now we have added schematically their functions and implications in some physiological and/or pathological conditions.
Round 2
Reviewer 1 Report
I have attached my revisions to this manuscript. My revisions mostly correct style and grammar. I suggest the authors incorporate my corrections. It will make the manuscript more readable. They should also work on it to make ithe mechanisms of CLIC channel function invarious diseases more clear and complete

Author Response
I have attached my revisions to this manuscript. My revisions mostly correct style and grammar. I suggest the authors incorporate my corrections. It will make the manuscript more readable
Thanks for your suggestions, they have been implemented in our manuscript.
They should also work on it to make ithe mechanisms of CLIC channel function invarious diseases more clear and complete.
We understand this point. However, we would like to focus on CLIC1 protein rather than deeply investigate CLIC2-6 functions. We followed the suggestion of Reviewer 2 about adding a schematical mention of CLIC2-6 proteins for a more wide picture without altering the focus of the work.